# EnvPool: A Highly Parallel Reinforcement Learning Environment Execution Engine

**Jiayi Weng**[†][*]  **Min Lin**[†]  **Shengyi Huang**[‡]  **Bo Liu**[§]
**Denys Makoviichuk**[♯]  **Viktor Makoviychuk**[△]  **Zichen Liu**[†□]
**Yufan Song**[◇]  **Ting Luo**[◇]  **Yukun Jiang**[◇]
**Zhongwen Xu**[†]  **Shuicheng Yan**[†]

[†]Sea AI Lab
[‡]Drexel University  [§]Peking University  [♯]Snap  [△]NVIDIA
[□]National University of Singapore  [◇]Carnegie Mellon University
trinkle23897@gmail.com, {linmin,xuzw,yansc}@sea.com

## Abstract

There has been significant progress in developing reinforcement learning (RL) training systems. Past works such as IMPALA, Apex, Seed RL, Sample Factory, and others, aim to improve the system's overall throughput. In this paper, we aim to address a common bottleneck in the RL training system, i.e., parallel environment execution, which is often the slowest part of the whole system but receives little attention. With a curated design for paralleling RL environments, we have improved the RL environment simulation speed across different hardware setups, ranging from a laptop and a modest workstation, to a high-end machine such as NVIDIA DGX-A100. On a high-end machine, EnvPool achieves one million frames per second for the environment execution on Atari environments and three million frames per second on MuJoCo environments. When running EnvPool on a laptop, the speed is $2.8\times$ that of the Python subprocess. Moreover, great compatibility with existing RL training libraries has been demonstrated in the open-sourced community, including CleanRL, rl_games, DeepMind Acme, etc. Finally, EnvPool allows researchers to iterate their ideas at a much faster pace and has great potential to become the de facto RL environment execution engine. Example runs show that it only takes five minutes to train agents to play Atari Pong and MuJoCo Ant on a laptop. EnvPool is open-sourced at https://github.com/sail-sg/envpool.

## 1 Introduction

Deep Reinforcement Learning (RL) has made remarkable progress in the past years. Notable achievements include Deep Q-Network (DQN) [22], AlphaGo [27, 29, 30, 31], AlphaStar [35], OpenAI Five [2], etc. Apart from the algorithmic innovations, the most significant improvements aimed at enhancing the training throughput for RL agents, such as leveraging the computation power of large-scale distributed systems and advanced AI chips like TPUs [16].

On the other hand, academic research has been accelerated dramatically by the shortened training time. For example, DQN takes eight days and 200 million frames to train an agent to play a single Atari game [22], while IMPALA [7] shortens this process to a few hours and Seed RL [6] continues to push the boundary of training throughput. This allows the researchers to perform iterations of their ideas at a much faster pace and benefits the research progress of the whole RL community.

---

[*]Currently at OpenAI. Detailed author contributions can be found in Appendix J.

36th Conference on Neural Information Processing Systems (NeurIPS 2022) Track on Datasets and Benchmarks.

Since training RL agents with high throughput offers important benefits, we focus on tackling a common bottleneck in the RL training system in this paper: parallel environment execution. To the best of our knowledge, it is often the slowest part of the whole system but has received little attention in previous research. The inference and learning with the agent policy network can easily leverage the experience and performance optimization techniques from other areas where deep learning has been applied, like computer vision and natural language processing, often conducted with accelerators like GPUs and TPUs. The unique technical difficulty in RL systems is the interaction between the agents and the environments. Unlike the typical setup in supervised learning performed on a fixed dataset, the RL systems must generate environment experiences at a very fast speed to fully leverage the highly parallel computation power of accelerators.

Our contribution is to optimize the environment execution for *general* RL environments, including video games and various applications of financial trading, recommendation systems, etc. The current method to run parallel environments is to execute the environment and pre-process the observation under Python multiprocessing. We accelerate the environment execution by implementing a general C++ threadpool-based executor engine that can run multiple environments in parallel. The well-established Python wrappers are optimized on the C++ side as well. The interactions between the agent and the environment are exposed by straightforward Python APIs as below.

```python
import envpool
import numpy as np

# make gym env
env = envpool.make("Pong-v5", env_type="gym", num_envs=100)
obs = env.reset()  # with shape (100, 4, 84, 84)
act = np.zeros(100, dtype=int)
obs, rew, done, info = env.step(act, env_id=np.arange(100))
# can get the next round env_id through info["env_id"]
```

The system is called EnvPool, a highly parallel reinforcement learning environment execution engine, where we support OpenAI gym APIs and DeepMind `dm_env` APIs. EnvPool has both synchronous and asynchronous execution modes, where the latter is rarely explored in the mainstream RL system implementation even though it has enormous potential. The currently supported environments on EnvPool include Atari [1], MuJoCo [33], DeepMind Control Suite [34], ViZDoom [17], classic RL environments like mountain car, cartpole [32], etc.

There are two groups of targeted users for EnvPool. One is RL researchers and practitioners who do not have to modify any parts of the RL environments. For example, researchers who would like to train an agent on Atari / MuJoCo tasks. They can use EnvPool just as OpenAI Gym, but faster. EnvPool intends to cover as many standard RL environments as possible in our GitHub repository. This group of users does not need to understand any internals of EnvPool, including any C++ code. They only work with the Python APIs (See Appendix A for comprehensive user APIs). The second group of "users" which we would like to call developers, are familiar with RL environment implementation (in C++) and would like to integrate their loved RL environments into EnvPool to speed up the environment execution. For this developer's group, we have provided extensive documentation on integrating a C++-based RL environment into EnvPool, including some straightforward examples (See Section 3 for more technical details).

Performance highlights of the EnvPool system include:

- With 256 CPU cores on an NVIDIA DGX-A100 machine, EnvPool achieves a simulation throughput of one million frames per second on Atari and three million physics steps per second on MuJoCo environments, which is $14.9\times$ / $19.2\times$ improvement over the current popular Python implementation [4] (i.e., 72K frames per second / 163K physics steps per second for the same hardware setup).

- On a laptop with 12 CPU cores, EnvPool obtains a speed $2.8\times$ of the Python implementation.

- When integrated with existing RL training libraries, example runs show that we can train agents to play Atari Pong and MuJoCo Ant on a laptop in five minutes.

- Sample efficiency is not sacrificed when replacing OpenAI gym with EnvPool and keeping the same experiment configuration. It is a pure speedup without cost.

## 2 Related Works

In this section, we review the existing RL environment execution component in the literature. Most implementations in RL systems use Python-level parallelization, e.g., For-loop or subprocess [4], in which we can easily run multiple environments and obtain the interaction experience in a batch. While the straightforward Python approaches are plugged easily with existing Python libraries and thus widely adopted, they are computationally inefficient compared to using a C++-level thread pool to execute the environments. The direct outcome of using inefficient environment parallelization is that more machines have to be used just for environment execution. Researchers build distributed systems like Ray [23] which allow easy distributed remote environment execution. Unfortunately, multiple third parties report an inefficient scale-up experience using Ray RLlib [19, 23] (cf. Figure 3 in [24]). This issue might be because, in a distributed setup, Ray and RLlib have to trade-off the communication costs with other components and are not specifically optimized for environment execution.

Sample Factory [24] focuses on optimizing the entire RL system for a single-machine setup instead of a distributed computing architecture. To achieve high throughput in the action sampling component, they introduce a sophisticated, fully asynchronous approach called Double-Buffered Sampling, which allows network forwarding and environment execution to run in parallel but on different subsets of the environments. Though having improved the overall throughput dramatically over other systems, the implementation complexity is high, and it is not a standalone component that can be plugged into other RL systems. Furthermore, Sample Factory sacrifices compatibility with a family of RL algorithms that can only work in synchronous mode to achieve high throughput. In contrast, EnvPool has both properties of high throughput and great compatibility with existing APIs and RL algorithms.

A few recent works, e.g., Brax [9], Isaac Gym [21], and WarpDrive [18], use accelerators like GPUs and TPUs for the environment engine. Due to the highly parallel nature of the accelerators, numerous environments can be executed simultaneously. The intrinsic drawback of this approach is that the environments must be purely compute-based, i.e., matrix operations so that they can be easily accelerated on GPUs and TPUs. They cannot handle general environments like common video games, e.g., Atari [1], ViZDoom [17], StarCraft II [35], and Dota 2 [2]. Moreover, in real-world applications, most scenarios cannot be converted into a pure compute-based simulation. Such a major drawback places the applications of this approach on a very limited spectrum.

The most relevant work to ours is the PodRacer architecture [11], which also implements the C++ batched environment interface and can be utilized to run general environments. However, their implementation only supports synchronous execution mode where PodRacer is operated on the whole set of environments at each timestep. The stepping will wait for the results returned by all environments and thus be slowed down significantly by the slowest single environment instance. The description of PodRacer architecture is specific to the TPU configuration. PodRacer is not open-sourced, and we cannot find many details on the concrete implementation. In contrast, EnvPool uses the asynchronous execution mode as a default to avoid slowing down due to any single environment instance. Moreover, it is not tied to any specific computing architectures. We have run EnvPool on both GPU machines and Cloud TPUs.

## 3 Methods

This section is largely intended for developers who are interested in the technical details of EnvPool and would like to contribute to the community. See Appendix A for detailed usage of EnvPool for RL researchers and practioners.

EnvPool contains three key components optimized in C++, the `ActionBufferQueue`, the `ThreadPool` and the `StateBufferQueue`. It uses pybind11 [15] to expose the user interface to Python. In this section, we start by providing an overview of the overall system architecture. We then illustrate the optimizations we made in the individual components. Finally, we briefly describe how to add new RL environments into EnvPool. Complete Python user APIs can be found in Appendix A.

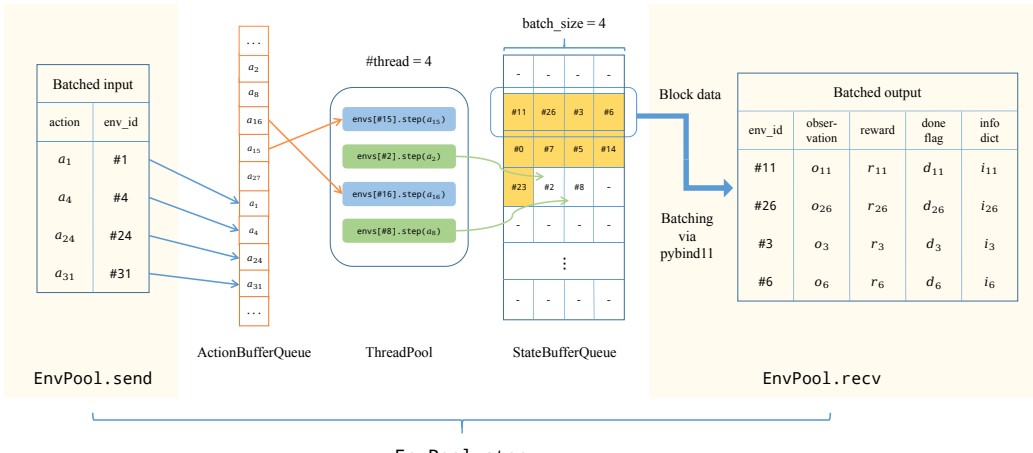

Figure 1: EnvPool System Overview

## 3.1 Overview

In the APIs of `gym` and `dm_env`, the central way to interact with RL environments is through the `step` function. The RL agent sends an action to the environment, which returns an observation. To increase the throughput of this interaction, the typical approach is to replicate it in multiple threads or processes. However, in systems that prioritize throughput (such as web services), the asynchronous event-driven pattern often achieves better overall throughput. This is because it avoids the context switching costs that arise in a simple multi-threaded setting.

EnvPool follows the asynchronous event-driven pattern visualized in Figure 1. Instead of providing a synchronous `step` function, in each interaction, EnvPool receives a batched action through the `send` function. The `send` function only puts these actions in the `ActionBufferQueue`, and returns immediately without waiting for environment execution. Independently, threads in the `ThreadPool` take action from the `ActionBufferQueue` and perform the corresponding environment execution. The execution result is then added to the `StateBufferQueue`, which is pre-allocated as blocks. A block in `StateBufferQueue` contains a fixed number (`batch_size` in the next section) of states. Once a block in the `StateBufferQueue` is filled with data, EnvPool will pack them into NumPy [10] arrays. The RL algorithm receives a batch of states by taking from the `StateBufferQueue` via the `recv` function. Details on the `ActionBufferQueue` and `StateBufferQueue` can be found in Appendix D.

A traditional `step` can be seen as consequent calls to `send` and `recv` with a single environment. However, separating `step` into `send/recv` provides more flexibility and opportunity for further optimization, e.g., they can be executed in different Python threads.

## 3.2 Synchronous vs. Asynchronous

A popular implementation of vectorized environments like `gym.vector_env` [4] executes all environments synchronously in the worker threads. We denote the number of environments `num_envs` as $N$. In each iteration, the input $N$ actions are first distributed to the corresponding environments, then wait for all $N$ environments to finish their executions. The RL agent will receive $N$ observation arrays and predict $N$ actions via forward pass. As shown in Figure 2 (a), the performance of the synchronous step is determined by the slowest environment execution time, and hence not efficient for scaling out.

Here we introduce a new concept `batch_size` in EnvPool's asynchronous `send/recv` execution. This idea was first proposed by Tianshou [36]. `batch_size` (denoted as $M$) is the batch size of environment outputs expected to be received by `recv`. As such, `batch_size` $M$ cannot be greater than `num_envs` $N$.

In each iteration, EnvPool only waits for the outputs of the first $M$ environment steps, and let other (unfinished) thread executions continue at the backend. Figure 2 (b) demonstrates this process with $N = 4$ and $M = 3$ in 4 threads. Compared with a synchronous step, asynchronous `send/recv` has a considerable advantage when the environment execution time has a large variance, which is a common when $N$ is large.

EnvPool can switch between synchronous mode and asynchronous mode by only specifying different `num_envs` and `batch_size`. In the asynchronous mode, `batch_size < num_envs`, the throughput is maximized. To switch to synchronous mode, we only need to set `num_envs = batch_size`, then consecutive calling `send/recv` is equivalent to synchronously stepping all the environments.

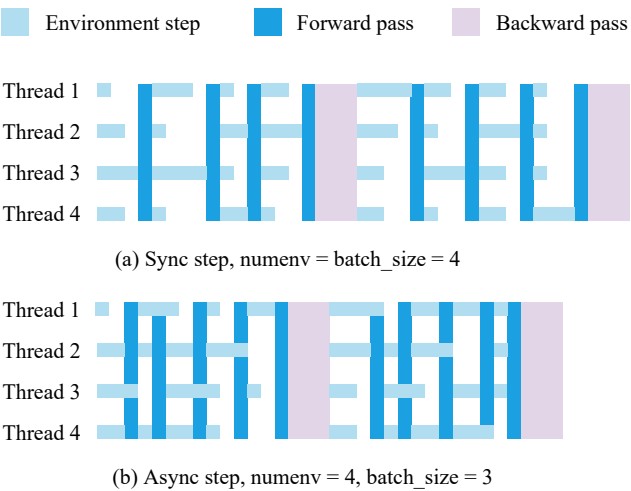

(a) Sync step, numenv = batch_size = 4

(b) Async step, numenv = 4, batch_size = 3

Figure 2: Synchronous step vs asynchronous step in EnvPool.

## 3.3 ThreadPool

ThreadPool [25] is a multi-thread executor implemented with `std::thread`. It maintains a fixed number of threads waiting for task execution without creating or destroying threads for short-term tasks. ThreadPool is designed with the following considerations:

- To minimize context switch overhead, the number of threads in ThreadPool is usually limited by the number of CPU cores.
- To further speed up ThreadPool execution, we can pin each thread to a pre-determined CPU core. This further reduces context switching and improves cache efficiency.
- We recommend setting `num_env` $N$ to be $2-3\times$ greater than the number of threads to keep the threads fully loaded when part of the envs are waiting to be consumed by the RL algorithm. On one hand, if we treat the environment execution time as a distribution, taking the $M$ environments with the shortest execution times can effectively avoid the long-tail problem; on the other hand, adding too many environments but keeping the `batch_size` unchanged may cause sample inefficiency or low-utilization of computational resources.

## 3.4 Adding New RL Environments

EnvPool is a highly extensible and developer-friendly platform for adding new reinforcement learning environments. The process is well-documented and straightforward for C++ developers[2].

First, developers need to implement the RL environment in a C++ header file. This involves defining the `EnvSpec` and the environment interface, which includes methods like `Reset`, `Step`, and `IsDone`. Next, they need to write a Bazel BUILD file to manage dependencies. They can then use these C++ source files to generate a dynamically linked binary, which can be instantiated in Python using

---

[2]https://envpool.readthedocs.io/en/latest/content/new_env.html

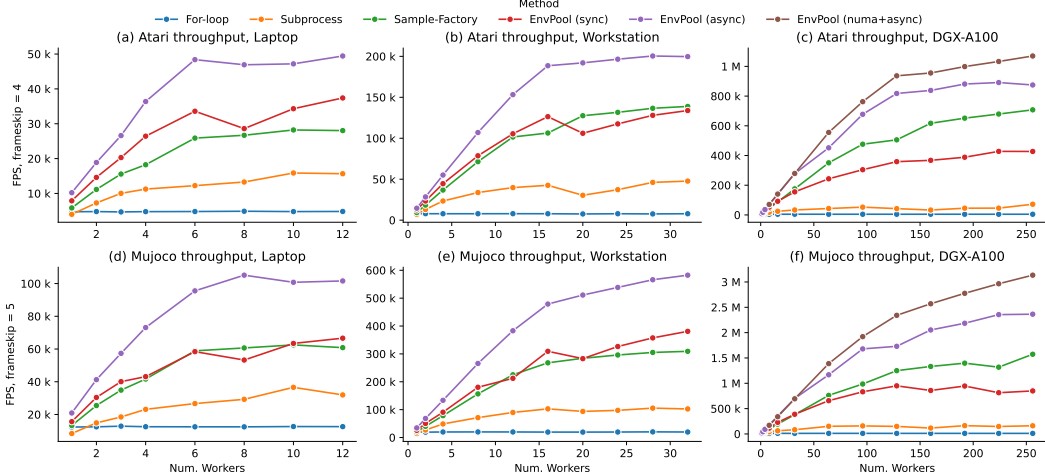

Figure 3: Simulation throughput in three machines with Atari and MuJoCo tasks.

pybind11. Finally, they need to register the environment in Python side and write rigorous unit tests for debugging.

One of the advantages of EnvPool is that adding new RL environments does not require a deep understanding of the core infrastructure. This makes it easy for developers to experiment with different environments and push the boundaries of RL research.

## 4 Experiments

Our experiments are divided into two parts. In the first part, we evaluate the simulation performance of the reinforcement learning environment execution engines, using randomly sampled actions as inputs. This isolated benchmark allows us to measure the performance of the engine component without the added complexity of agent policy network inference and learning.

In the second part of our experiments, we assess the impact of using EnvPool with existing RL training frameworks. We test EnvPool with CleanRL [14], rl_games [20], and DeepMind's Acme framework [12] to see how it can improve overall training performance.

Overall, our experiments demonstrate the value of EnvPool as a tool for improving the efficiency and scalability of RL research. By optimizing the simulation performance of RL environments, EnvPool allows researchers to train agents more quickly and effectively.

### 4.1 Pure Environment Simulation

We first evaluate the performance of EnvPool against a set of established baselines on the RL environment execution component. Three hardware setups are used for the benchmark: the Laptop setting, the Workstation setting, and the NVIDIA DGX-A100 setting. Detailed CPU types and specifications can be found in Appendix B.

The laptop has 12 CPU cores, and the workstation has 32 CPU cores. Evaluating EnvPool on these two configurations can demonstrate its effectiveness with small-scale experiments. An NVIDIA DGX-A100 has 256 CPU cores with 8 NUMA nodes. Note that running multi-processing on each NUMA node not only makes the memory closer to the processor but also reduces the thread contention on the `ActionBufferQueue`.

As for the RL environments, we experiment on two of the most used RL benchmarks, namely, Atari [1] with Pong and MuJoCo [33] with Ant. In the experiments of pure environment simulation, we obtain a randomly sampled action based on the action space definition of the games and send the actions to the environment executors. The number of frames per second is measured with a mean of

Table 1: Numeric results for benchmarking.

| System Configuration | Laptop | | Workstation | | DGX-A100 | |
|---|---|---|---|---|---|---|
| Method \ Env (FPS) | Atari | MuJoCo | Atari | MuJoCo | Atari | MuJoCo |
| For-loop | 4,893 | 12,861 | 7,914 | 20,298 | 4,640 | 11,569 |
| Subprocess | 15,863 | 36,586 | 47,699 | 105,432 | 71,943 | 163,656 |
| Sample-Factory | 28,216 | 62,510 | 138,847 | 309,264 | 707,494 | 1,573,262 |
| EnvPool (sync) | 37,396 | 66,622 | 133,824 | 380,950 | 427,851 | 949,787 |
| EnvPool (async) | **49,439** | **105,126** | **200,428** | **582,446** | 891,286 | 2,363,864 |
| EnvPool (numa+async) | / | / | / | / | **1,069,922** | **3,134,287** |

50K iterations, where the Atari frame numbers follow the practice of IMPALA [7] and Seed RL [6] with frameskip set to 4, and MuJoCo sub-step numbers set to 5.

We compare several concrete implementations extensively, which are described below. Among them, Subprocess is the most popular implementation currently and, to the best of our knowledge, Sample Factory is the best performing general RL environment execution engine at the time of publication.

- For-loop: execute all environment steps synchronously within only one thread;
- Subprocess [4]: execute all environment steps synchronously with shared memory and multiple processes.
- Sample Factory [24]: pure asynchronous step with a given number of worker threads; we pick the best performance over various `num_envs` per worker.
- EnvPool (sync): synchronous step execution in EnvPool.
- EnvPool (async): asynchronous step execution in EnvPool; given several worker threads for `batch_size`, pick the best performance over various `num_envs`.
- EnvPool (numa+async): use all NUMA nodes, each launches EnvPool individually with asynchronous execution to see the best performance of EnvPool.

To demonstrate the scalability of the above methods, we conduct experiments using various numbers of workers for the RL environment execution. The experiment setup ranges from a couple of workers (e.g., 4 cores) to using all the CPU cores in the machine (e.g., 256 cores).

Our EnvPool system outperforms all of the strong baselines with significant margins on all hardware setups of the Laptop, Workstation, and DGX-A100 (Figure 3 and Table 1). The most popular Subprocess implementation has extremely poor scalability with an almost flat curve. This indicates a small improvement in throughput with the increased number of workers and CPUs. The poor scaling performance of Python-based parallel execution confirms the motivation of our proposed solution.

The second important conclusion is that, even if we use a single environment in EnvPool, we can get a free $\sim 2\times$ speedup. Complete benchmarks on Atari, MuJoCo, and DeepMind Control can be found in Appendix C.

The third observation is that synchronous modes have significant performance disadvantages against asynchronous systems. This is because the throughput of the synchronous mode execution is determined by the slowest single environment instance, where the stepping time for each environment instance may vary considerably, especially when there is a large number of environments.

## 4.2 End-to-end Agent Training

In this work, we demonstrate successful integration of EnvPool into three different open-sourced RL libraries. EnvPool can serve as a drop-in replacement of the vectorized environments in many deep RL libraries and reduce the training wall time without sacrificing sample efficiency. The integration with training libraries has been straightforward due to compatibility with existing environment APIs. These example runs were performed by practitioners and researchers themselves, reflecting realistic use cases (e.g., using their machines and their preferred training libraries) in the community.

The full results cover a wide range of combinations to demonstrate the general improvement on different setups, including different training libraries (e.g., PyTorch-based, JAX-based), RL environ-

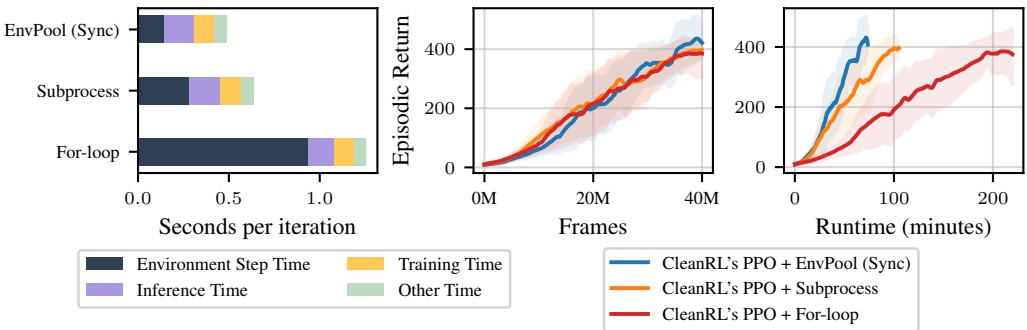

Figure 4: A profile of CleanRL's PPO in the Atari game Breakout using $N = 8$.

ments (e.g., Atari, MuJoCo), machines (e.g., laptops, TPU VMs). We present the main findings in the following paragraphs, where results are aggregated over five random seeds, the learning curves are smoothed by a moving average of window size 10, and the shaded region of the learning curves represents one standard deviation of episodic returns. The complete configurations and results can be found in Appendix F. Note that the hardware specifications of these experiments are different thus readers should *not* compare training speeds across different training libraries.

**How much time does parallel environment execution take?** As a case study, we profile CleanRL's PPO in Atari games with three parallelization paradigms — For-loop, Subprocess, and EnvPool (Sync). CleanRL's PPO is empirically shown to be equivalent to `openai/baselines`' PPO [13], and we use the same PPO hyperparameters used in the original PPO paper [28], which uses $N = 8$. Specifically, we measure the following times per iteration over 9,765 iterations of rollout and training:

1. Environment Step Time: the time spent on `env.step(act)` (i.e., stepping 8 actions in 8 environments and returning a batch of 8 observations, rewards, dones, and infos).

2. Inference Time: the time spent on computing actions, log probabilities, values, and the entropy.

3. Training Time: the time spent doing forward and backward passes.

4. Other Time: the time spent on other procedures (e.g., storage, moving data between GPU and CPU, writing metrics, etc).

The results are presented in Figure 4. We clearly see that Environment Step Time is a significant bottleneck under Python-level parallelization, and EnvPool (Sync) ameliorates this bottleneck. As a result, the end-to-end training time decreased from 200 minutes (CleanRL's PPO + For-loop) to approximately 73 minutes (CleanRL's PPO + EnvPool (Sync)) while maintaining sample efficiency. Furthermore, we should expect a further speed up with a larger number of $N$, such as 32 or 64 when using EnvPool compared to other parallelization paradigms.

Note that the above case study could look drastically different based on 1) the number of environments (e.g., $N = 8$, 32, or 64); 2) the type of environments (e.g., MuJoCo or Atari); 3) the learning parameters (e.g., the number of mini batches used); and 4) the used deep RL library. Even so, what is important is that EnvPool can speed up the Environment Step Time in the overall training system with $N > 1$.

**Easy integration with popular deep RL libraries.** Since many deep RL libraries utilize vectorized environments with some form of parallel environment executors, integrating EnvPool to them is straightfoward. In this work, we additionally present successful integration with rl_games [20] and DeepMind Acme [12]. For example, Figure 5 shows multiple folds of wall-time training speed improvement in rl_games when using EnvPool versus its default parallel environment executor built on top of Ray [23]. Further results on Acme can be found in Appendix F.

**High throughput training.** Additionally, we can search for an alternative set of hyperparameters that better leverage EnvPool's throughput. For example, in MuJoCo, Schulman et al. [28] use a single simulation environment and let PPO use 32 mini-batches and 10 update epochs, which results in 320 gradient updates per batch of rollout data. This results in stale data after the first gradient update

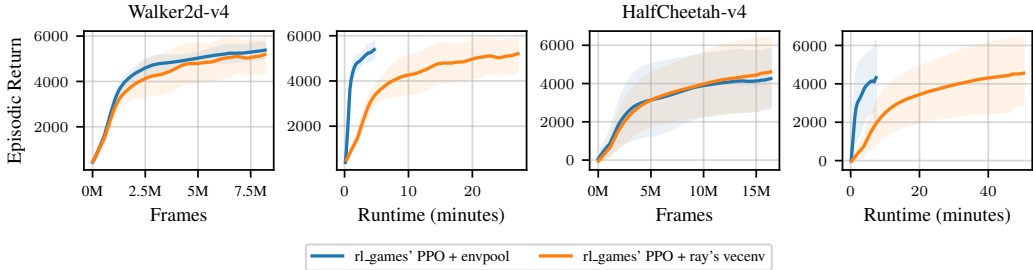

Figure 5: rl_games example runs with Ray and with EnvPool, using the same number of parallel environments $N = 64$

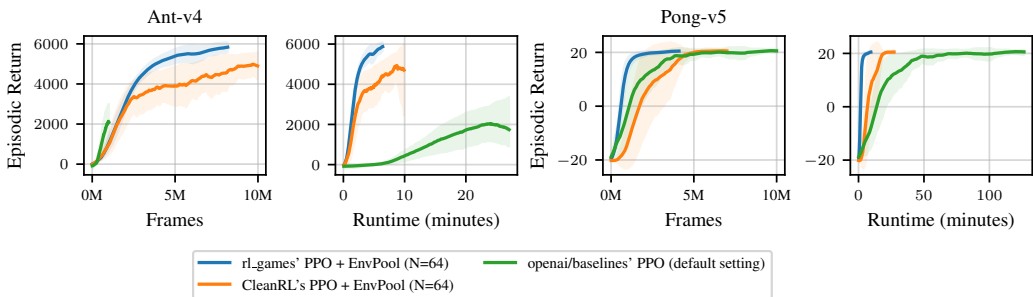

Figure 6: rl_games and CleanRL example runs with $N = 64$ and tuned parameters compared to `openai/baselines`' PPO which by default uses $N = 1$ for MuJoCo and $N = 8$ for Atari experiments [28].

(i.e., the optimized policy is newer than the behavior policy that was used to collect the rollout data; see [13] for more details). To reduce the stale data, we could use a higher number of simulation environments, such as $N = 64$ and fewer mini-batches and update epochs.

For example, in Figure 6 example runs, rl_games PPO can solve Ant in five minutes of training time, while OpenAI baselines' PPO can only get to score 2,000 in 20 minutes. Such a significant speed up on a laptop-level machine benefits researchers in terms of a rapid turnaround time of their experiments. We note that a drop in sample efficiency is observed in these runs. Similar training speedup observations can be drawn from the example run in Figure 6. OpenAI baselines' PPO requires training of 100 minutes to solve Atari Pong, while rl_games can tackle it within a fraction of time of the baseline, e.g., five minutes.

## 5 Future Work

**Completeness**: In this publication, we have only included RL environments with Atari [1], Mu-JoCo [33], DeepMind Control Suite [34], ViZDoom [17], and classic ones like mountain car, cartpole, etc. We intend to expand our pool of supported environments to cover more research use cases, e.g., grid worlds that are easily customized to research [5]. On the multi-agent environments, we have implemented ViZDoom [17] and welcome the community to add even more environments including Google Research Football, MuJoCo Soccer Environment, etc.

**Cross-platform Support**: The EnvPool intends to support extra operating systems, such as MacOS and Windows.

**User friendliness**: We intend to create a template repository to help customized environment integration into EnvPool easier, so that users can develop their own environment without having to work under EnvPool's code base while still having access to register the self-written environment with EnvPool and use the `make` function to create it.

**Distributed Use Case**: The EnvPool experiments in the paper have been performed on single machines. The same APIs can be extended to a distributed use case with remote execution of the environments using techniques like gRPC. The logic behind the environment execution is still hidden from the researchers but only the machines used to collect data will be at a much larger scale.

**Research Directions**: With such a high throughput of data generation, the research paradigm can be shifted to large-batch training to better leverage a large amount of generated data. There are no counterparts as successful in computer vision and natural language processing fields, where large-batch training leads to stable and faster convergence. An issue induced by faster environment execution would be severe off-policyness. Better off-policy RL algorithms are required to reveal the full power of the system. Our proposed system also brings many new opportunities. For example, more accurate value estimations may be achieved by applying a large amount of parallel sampling, rollouts, and search.

# 6 Conclusion

In this work, we have introduced a highly parallel reinforcement learning environment execution engine EnvPool, which significantly outperforms existing environment executors. With a curated design dedicated to the RL use case, we leverage techniques of a general asynchronous execution model, implemented with a C++ thread pool on the environment execution. For data organization and outputting batch-wise observations, we designed BufferQueue tailored for the RL environments. We conduct an extensive study with various setups to demonstrate the scale-up ability of the proposed system and compare it with both the most popular implementation gym and highly optimized system Sample Factory. The conclusions hold for both Atari and MuJoCo, two of the most popular RL benchmark environments. In addition, we have demonstrated significant improvements in existing RL training libraries' speed when integrated with EnvPool in a wide variety of setups, including different machines, different RL environments, different RL algorithms, etc. On laptops with a GPU, we managed to train Atari Pong and MuJoCo Ant in five minutes, accelerating the development and iteration for researchers and practitioners. However, some limitations remain, for example, EnvPool cannot speed up RL environments originally written in Python and therefore developers have to translate each existing environment into C++. We hope that EnvPool will become a core component of modern RL infrastructures, providing easy speedup and high-throughput environment experiences for RL training systems.

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
