# OpenReview forum: "EnvPool: A Highly Parallel Reinforcement Learning Environment Execution Engine"
_NeurIPS.cc/2022/Track/Datasets_and_Benchmarks — NeurIPS 2022 Datasets and Benchmarks _

### Official Review · Reviewer_CyiY · 2022-07-21
**EnvPool addresses a significant bottleneck for RL**

**Rating:** 7
**Confidence:** 4
**Clarity:** Yes, the paper is clearly organized a…

**Strengths:**

The authors have demonstrated significant speedups in some of the most common environments used for reinforcement learning: Atari and MuJoCo. Completing training on these environments can be a significant barrier to research: researchers may need to run on many different environments over many different seeds. Increasing the speed of training may make RL research accessible to more researchers and improve the field as a whole.

The generic interface provided by EnvPool makes adoption simple for those already using environments supported by EnvPool. For new environments, EnvPool provides significant documentation on developing compatibility.

**Weaknesses:**

Because some engineering effort is required to adapt new environments to EnvPool, it may not be useful to all researchers in the field. In addition, it is unclear how much improvement EnvPool will provide to other types of environments, in particular for more GPU intensive environments.

Many RL algorithms are bound by training compute rather than sampling time, and in this case EnvPool also may not provide significant benefit.

**Additional Feedback:**

None

**Correctness:**

Yes, the claims appear correct. The evaluation methods use multiple trials and clearly state their limitations.

**Documentation:**

The EnvPool code is well documented and is actively supported on GitHub. I was able to replicate the results of the pure environment benchmarking using documented code.

**Relation To Prior Work:**

Yes, comparisons to prior work such as SampleFactory and Ray are discussed in sufficient detail.

**Summary And Contributions:**

EnvPool speeds up execution of Reinforcement Learning environments by running them in parallel. EnvPool uses C++ threading for environment execution but provides a standard python interface for learning at the batch level. EnvPool improves upon other implementations by allowing either synchronous or asynchronous batching. The authors demonstrate speedups in both raw environment frame rates and algorithmic training time without loss of sample efficiency.

---

> ### Author Response · Authors · 2022-08-10
> **Response to Reviewer CyiY**
>
> We thank the reviewer for their constructive feedback.
>
> On the usefulness for researchers in the RL field, we've provided clarification in [Common questions](https://openreview.net/forum?id=BubxnHpuMbG&noteId=g5xm5Drohi6). Hope that our answer can address your concerns.
>
> > Q:  Many RL algorithms are bound by training compute rather than sampling time, and in this case EnvPool also may not provide significant benefit.
>
> A: It is the other way around. We usually use CPUs to get RL training samples from environments and GPU(s) to inference and train RL policies. In a common use case, you should notice that GPU utilization is always low (e.g., under 10%) when training with on-policy RL algorithms. The aforementioned scenario can be found during general RL training. Because of the reason that policy networks are typically small (e.g., two layers of ConvNets in Atari, three layers of MLP for Mujoco tasks) in RL implementation, inference speed is so fast that sometimes it may wait for a CPU sampler to generate samples (that’s the whole idea of ‘vec_env’ originated from). Also, it is costly to obtain samples interacting with environments, you may find evidence in some notable large-scale RL training papers (i.e. OpenAI Five[1] 256 GPUs vs 128,000 CPUs; Tencent JueWu [2] 1,064GPUs vs 600,000 CPUs).
>
> We also provide an extra example to prove our point, which you can find in our revision, Sec 4.2 End-to-end Agent Training and Figure 4, where we plot the time on environment stepping, policy inference, training, etc. We can tell very clearly from Figure 4 that environment step time is the dominant one and slowed down the overall system.
>
> [1] Berner, C., Brockman, G., Chan, B., Cheung, V., Dębiak, P., Dennison, C., ... & Zhang, S. (2019). Dota 2 with large scale deep reinforcement learning. arXiv preprint arXiv:1912.06680
>
> [2] Ye, D., Liu, Z., Sun, M., Shi, B., Zhao, P., Wu, H., ... & Huang, L. (2020, April). Mastering complex control in moba games with deep reinforcement learning. In Proceedings of the AAAI Conference on Artificial Intelligence (Vol. 34, No. 04, pp. 6672-6679).

---

> > ### Comment · Reviewer_CyiY · 2022-08-25
> > **Added section on End-to-end Agent Training helps**
> >
> > Thanks to the authors for the thoughtful response and additional section. Does EnvPool reduce the compute required for sampling the environment, or only the latency?
> >
> > My comment was intended to highlight that the speedup gained from switching a system from another parallelization scheme like subprocessing to EnvPool is not 3x (as it would be if the system were entirely limited by sampling the environment), but closer to 30%. This is nicely demonstrated by the new Figure 4.
> >
> > I appreciate the suggestion to take advantage of lower latency to keep more on policy steps, described in the expanded high throughput training section. I believe that this paper remains strong and is a valuable contribution to the field.

---

### Official Review · Reviewer_KzcX · 2022-07-22
**Review: Speeding up existing environments is good contribution to the RL community**

**Rating:** 6
**Confidence:** 4

**Strengths:**

- EnvPool tries to solve the execution-time bottleneck in RL research. This seems to be an important contribution to the community, since the researcher can run the try-and-error cycle faster.
- EnvPool can train Ant in 5 min, which seems comparable to hardware-accelerated simulators, such as Brax or IsaacGym.
- Although Brax and IsaacGym have their own eco-system of the environments, EnvPool supports existing environments widely used in previous RL research, which is beneficial for benchmarking.

**Weaknesses:**

- This framework is written in C++, not in Python, a standard/widely-used ML community programming language. This may limit the actual developers/users from RL research communities (e.g. Brax is all written in Python library: Jax. IsaacGym supports rich Python bindings.).
- If we want to use self-made environments, we must design additional environment wrappers in C++ per novel environments. This might be a bit costly, compared to the existing parallel engine (Subprocess, Sample Factory, etc), which can be used in common across environments.
- As mentioned in Section 5, it might be unclear whether EnvPool is superior to the existing large-scale RL works (i.e. highly-distributed IMPALA, Seed RL)

**Additional Feedback:**

I feel the development of EnvPool seems an important contribution to the community. One concern is sustainability since the users should write a C++ wrapper, not an ML-friendly programing language like Python. This may limit the number of users/developers.

**Clarity:**

- This paper is well-structured and easy to follow.
- It may be better to fix the caption of Figure 6 and Figure 7. Does blue *rl_games (tuned)* also use EnvPool, right?

**Correctness:**

- The comparison and benchmark of execution time/performance seem well-designed and fair.

**Documentation:**

- I checked the github repo (https://github.com/sail-sg/envpool) and documentation (https://envpool.readthedocs.io/en/latest/index.html). These seem actively developed, maintained, and well-organized for users.
- https://github.com/sail-sg/envpool has example usage with existing RL algorithm libraries (Stable-Baselines3, Tianshou, ACME, CleanRL, or rl_games).

**Relation To Prior Work:**

- This paper compares the proposed EnvPool with a variety of existing parallelization engines (e.g. Ray, Subprocess, Sample Factory) on a variety of environments (Atari, MuJoCo) and RL frameworks (baselines, rl_games, clean rl, acme).
- As mentioned in `Summary And Contributions`, EnvPool seems to depend less on high-end GPU/TPUs and can support existing environments widely used in previous works, compared to recent massively-parallelized hardware-accelerated simulators, such as Brax or IsaacGym.

**Summary And Contributions:**

This paper proposes *EnvPool*, the C++-based framework to accelerate OpenAI Gym-style RL environment execution with efficient parallelization. EnvPool has both Synchronous and Asynchronous communication to the thread, and especially Asynchronous EnvPool significantly speeds up FPS of RL training in any kind of environment (MuJoCo/Atari, etc) on any computers/servers (Laptop, Workstation, DGX-A100). For example, PPO on EnvPool can train MuJoCo Walker2d within 10 minutes and Ant within 5 minutes, which seems faster than the existing parallelization frameworks. Furthermore, in contrast to recent hardware-accelerated simulators (Brax, IsaacGym), EnvPool seems to depend on GPU/TPUs less and can support existing environments widely used in previous RL research.

---

> ### Author Response · Authors · 2022-08-09
> **Response to Reviewer KzcX**
>
> We thank the reviewer for their valuable feedback.
>
> For the concern about C++ and overheads to integrate RL environments into EnvPool, we've addressed in the [Common questions asked by a few reviewers](https://openreview.net/forum?id=BubxnHpuMbG&noteId=g5xm5Drohi6). We clarify that typical RL researchers and users would not need to understand any C++ code. They can just use our Python APIs to get the same experience as what they have with OpenAI gym APIs.
>
> > Q: It might be unclear whether EnvPool is superior to the existing large-scale RL works
>
> The philosophy of EnvPool is not to create an RL framework, but to provide a standalone piece that can be used in whatever way the user would like to. As shown in our paper, we’ve worked with authors and users of different RL training libraries, including CleanRL, rl_games, acme, etc, where they can empower their favorite training systems with the EnvPool executor.
>
> Meanwhile, EnvPool targets small-scale research-friendly computation settings, e.g., a laptop, and can achieve the same level of performance compared to previous large-scale RL works, e.g., IMPALA and SeedRL.

---

### Official Review · Reviewer_jfBd · 2022-07-23
**A system for paralleling RL environments by using a C++-level thread pool.**

**Rating:** 7
**Confidence:** 4
**Correctness:** The project is solid and the experime…
**Clarity:** It is easy to read.

**Strengths:**

The rollout efficiency is a huge concern for RL researchers, especially those trying RL on complex control tasks, *i.e.* Robotics. Due to the inefficient simulation and frequent communication between GPU and CPU, it usually takes days or even weeks to tune parameters and verify ideas. Although recent works such as IsaacGym achieve parallel GPU simulation to address inefficient simulation and eliminate communication costs, there are still a large number of environments/simulators that are built to run on CPU. Therefore, it is **valuable** to develop tools, like EnvPool for boosting the rollout efficiency of CPU-based simulation.

This project is highly developed with a well-constructed codebase and comprehensive documents. Currently, it already supports many commonly used RL environments, like Mujoco and Atari. The authors conduct experiments in these environments. It turns out that EnvPool outperforms other paralleling schemes by a large margin.




**Weaknesses:**

The weakest part of EnvPool is that it is difficult to be applied to custom environments/simulators. Although EnvPool shows astonishing performance on Gym and Mujoco tasks, for researchers using these environments, the rollout efficiency is not a sticky problem. As I mentioned before, there are a lot of custom environments that require powerful paralleling schemes. Therefore, it is more important to make the combination between EnvPool and custom environments easy than to officially support common environments.

Actually, I would like to try it on our in-house simulator which is built on the Bullet Physics engine, and report some results. However, the process to add a custom environment is too complex, so I give up. Also, I found that it currently only supports the Linux platform. Is there any plan to make it available on other platforms, like Windows? There are some cross-discipline researchers whose simulator is built on Windows. Therefore, I suggest Windows support, at least, should be added.

Overall, I really like this valuable work. If the authors could describe their future developing plan or make some commitments to address the issues (especially the compatibility issue) in the revision, I think it deserves a higher score.

**Additional Feedback:**

N/A

**Documentation:**

The document looks complete.

**Ethics:**

No ethical issue.

**Relation To Prior Work:**

The survey is comprehensive.

**Summary And Contributions:**

This system aims to reduce the time to collect transitions for on-policy RL methods, like PPO. With the rapid training enabled by EnvPool, researchers can prototype new ideas faster, thus accelerating the progress of RL research. Currently, most paralleling rollout environment wrappers suffer from inefficient Python thread execution. EnvPool solves this problem by building the paralleling system with C++, which can access the computation resources in a more straightforward manner than Python. The experiment shows that EnvPool can greatly reduce the training time, compared to Ray and  Sample Factory, especially in async mode. The code is released, and tutorials and instructions are well documented.

---

> ### Author Response · Authors · 2022-08-09
> **Committed to supporting more platforms and addressing compatibility issues**
>
> We thank the reviewer for their constructive feedback and recognition of our work.
>
> > Q: Is there any plan to make it available on other platforms, like Windows?
>
> A: Yes, we’ve received requests to make EnvPool generally available on more platforms, see [Issue #168](https://github.com/sail-sg/envpool/issues/168). It is on the roadmap for EnvPool v0.7.0 release, which will be due on Oct 3, 2022. Detailed list can be found in https://github.com/sail-sg/envpool/milestone/3
>
> > Q: Authors could describe their future developing plan or make some commitments to address the issues (especially the compatibility issue).
>
> A: We provide a detailed future developing plan in Section 5 future work. We have added more plans regarding EnvPool v0.7.0 release in the revised version. Also as you can tell from the past GitHub activities, we’re actively maintaining and developing EnvPool, committed to providing a good user experience for researchers and users.

---

### Official Review · Reviewer_hVMa · 2022-07-26
**A RL system with highly parallel reinforcement learning environments.**

**Rating:** 7
**Confidence:** 4
**Correctness:** YES, but mostly correct.
**Clarity:** YES.

**Strengths:**

The benchmark can be suitable for different RL environments, and the issue of paralleling RL environments is addressed well.

**Weaknesses:**

1. The authors claim in Section 1 that " we focus on tackling a common bottleneck in the RL training system in this paper: parallel environment execution. To the best of our knowledge, it is often the slowest part of the whole system but has received little  attention in the previous literature. ". Consequently, I think the in-depth analysis of the impact of environment execution throughput on overall training speed should be supplemented.

2. In Section 1, the author claims that EnvPool is 14.9/19.2 times faster than the “Python implementation [3]” , but do not state how the Python implementation utilizes hardware devices. Reference [3] is only the simulation environment OpenAI gym, not a reinforcement learning training framework.

3. Compared with the related work of using GPU for environment parallel acceleration mentioned in Section 2, the speed of this research utilizing CPU for environment parallel acceleration should be much slower. The proposed system also requires low-level editing of the environment (converting it to C++ code), which is also limited in scope.

4. In Section 3 describing the specific methods, the proposed parallelization acceleration methods are basically generally adopted, and lack novelty and contribution.

5. In the experiments in Section IV, experimental comparisons with state-of-the-art methods using GPU acceleration are lacking.

6. Based on Figure 3 and Table 1 in Section 4.1, the sample throughput of Sample Factory is roughly comparable to that of the proposed EnvPool system. Howerer, why is Sample Factory not included in the overall training comparison in Section 4.2? Why did the authors switch to the RAY system that was reported to be inefficient for comparison? In my experience, doubling the experience sampling speed may not result in a significant improvement in overall training speed.

**Additional Feedback:**

More environments may need to be integrated, such as grid worlds.

**Documentation:**

YES.

**Ethics:**

From my side, didn't find any ethical concerns.

**Relation To Prior Work:**

YES.

**Summary And Contributions:**

The EnvPool system proposed in this paper achieves improved throughput in some cases by programming the environment in C++.

---

> ### Author Response · Authors · 2022-08-10
> **Response to Reviewer hVMa**
>
> We thank the reviewer for their careful evaluation and helpful feedback!
>
> > Q: Consequently, I think the in-depth analysis of the impact of environment execution throughput on overall training speed should be supplemented.
>
> A: This is a good point! Thank you for suggesting it. We’ve added a section for the in-depth analysis on Page 7 to Page 8, where we analyze the time spent on environment stepping, inference, training, and other overheads. As we can see from Figure 4 clearly, the slow environment stepping is the bottleneck of the overall training system.
>
> > Q: the author claims that EnvPool is 14.9/19.2 times faster than the “Python implementation [3]” , but do not state how the Python implementation utilizes hardware devices
>
> A: Thanks for pointing out this issue. The Python implementation can only use multiple CPUs via Python subprocess. The hardware requirements are the same as EnvPool. We will update this information in our paper accordingly.
>
> > Q: the speed of this research utilizing CPU for environment parallel acceleration should be much slower. The proposed system also requires low-level editing of the environment (converting it to C++ code), which is also limited in scope
>
> A: We’ve provided our answers in the [Common questions](https://openreview.net/forum?id=BubxnHpuMbG&noteId=g5xm5Drohi6) part. Please refer to that. Hope that it addresses your concerns.
>
> > Q: Experimental comparisons with state-of-the-art methods using GPU acceleration are lacking.
>
> A: We’ve addressed the concerns in [Common questions](https://openreview.net/forum?id=BubxnHpuMbG&noteId=g5xm5Drohi6).
>
> > Q: Why is Sample Factory not included in the overall training comparison in Section 4.2?
>
> A: Adding SF (Sample Factory) to the overall training comparison is considerably more difficult than you thought for two reasons.
> - SF1 does not really support running end-to-end training with Atari — the original paper only shows Atari throughput (Figure 3 in Petrenko et al.) and the author suggested they did not train Atari policies (see https://github.com/alex-petrenko/sample-factory/issues/51). Besides, there was no documentation on how to properly use the Atari wrappers, thus making it challenging to run end-to-end experiments and make fair comparison with our experiments which did use the Atari wrappers;
> - The new SF2 is not ready. The SF team is working on releasing SF2 (Sample Factory 2), which does support running Atari end-to-end (which we wish to compare). However, this effort is still ongoing (see https://github.com/alex-petrenko/sample-factory/issues/158 ) In fact, our co-author Shengyi is helping the SF team to match openai/baseline’s PPO’s sample efficiency in Atari experiments.

---

### Official Review · Reviewer_DBB4 · 2022-07-27
**The paper presents a helpful tool to accelerate research in RL, though its usefulness is limited a bit by the complex way of integrating new custom environments.**

**Rating:** 7
**Confidence:** 4

**Strengths:**

- Parallelizing arbitrary RL environments has the potential to speed up research in many directions and enable researchers to try more of their ideas in a shorter span of time.
- Focussing on the single machine use-case improves the situation for the usual RL researcher without a large cluster at hand.
- The API is simple to use and can be integrated easily into existing RL training setups.
- The most relevant environments are implemented and can be used off the shelf.
- The environment parallelization can also be optimized via XLA which is especially useful for jax-based agents.

**Weaknesses:**

- Adding an environment requires the C++ build setup and several steps, which is not as easy as the authors make it sound in the paper. However, there is sufficient documentation to guide the researchers through the process.
- The XLA optimization feature is not evaluated but would be really interesting as well for the paper.

**Additional Feedback:**

- The authors should note, what the shaded area in e.g. figure 4 is.

**Clarity:**

- The paper is well written and the structure makes it easy to follow.

**Correctness:**

- The paper's presentation and evaluation of the tool is sound and highlights the characteristics of it well.

**Documentation:**

- The documentation is quite detailed, but could include a note for more complex environment observation spaces such as `Dict` or other combined spaces.

**Ethics:**

- There are no ethical concerns that I know of.

**Relation To Prior Work:**

- Other approaches to parallelize environments as well as their shortcomings are described in enough detail.

**Summary And Contributions:**

- The authors present `envpool`, a C++ tool with python bindings to parallelize RL environments.
- The tool is able to wrap arbitrary environments that do not have to be parallelizable using matrix operations.
- The environments can be stepped in a synchronous way and also asynchronously to speed up the evaluation even further.
- The documentation is well developed and readable.

---

> ### Author Response · Authors · 2022-08-10
> **Response to Reviewer DBB4**
>
> We thank the reviewer for their recognition of our work.
>
> On the question of C++ build setup, we've addressed in [Common questions](https://openreview.net/forum?id=BubxnHpuMbG&noteId=g5xm5Drohi6), please refer to the answer there. It should clarify on two different targeted groups.
>
> > Q: The XLA optimization feature is not evaluated
>
> Thanks for spotting the XLA feature from our GitHub, we didn't mention it in the manuscript since it's still in the early stage and under active development.
> We have had some promising results (see https://imgur.com/a/WCotlJv) that show envpool’s XLA interface could offer a **70%** speed up compared to the `PPO + PyTorch + EnvPool` variant presented in Figure 4 at no cost of sample efficiency (i.e., `PPO + JAX + EnvPool’s XLA` takes ~42 mins to finish and `PPO + Torch + EnvPool` takes 72 mins.)
> That said, the development is still ongoing but we might be able to fit in the camera-ready version. The CleanRL’s `PPO + JAX + XLA` code has not been cleaned up yet, and there are subtle issues. For example, using the XLA interface makes reporting raw episodic returns difficult — XLA requires everything to have fixed shapes but one rollout could produce a variable number of episodic returns. So the current prototype only reports the average episodic return, which makes it hard to compare exactly against prior results.
>
> We're happy to provide more details if you're curious!

---

> > ### Comment · Reviewer_DBB4 · 2022-08-13
> > **Response**
> >
> > I thank the authors for their detailed response and would be grateful to get more details about the XLA integration. I think that this topic will become increasingly important in the future and seeing the pitfalls and difficulties would be of great value.

---

### Official Review · Reviewer_9AXU · 2022-07-28
**Useful Software with Questionable Experiments**

**Rating:** 7
**Confidence:** 3

**Strengths:**

EnvPool is the first parallel environment execution engine which allows for asynchronous environment execution. This allows EnvPool to significantly increase the speed of environment execution in parallel.  Due to this parallel environment execution, training times for RL algorithms can be reduced, increasing the pace at which research progresses.



**Weaknesses:**

EnvPool does not seem to be a dataset or a benchmark, but rather software to improve the efficiency of working with benchmarks. The NeurIPS datasets and benchmarks track is for datasets/benchmarks and a forum for discussion on how to improve the development of datasets/benchmarks (see https://neurips.cc/Conferences/2022/CallForDatasetsBenchmarks).  Although I have no doubt about the utility of EnvPool, I am unsure whether EnvPool meets these criteria.


**Additional Feedback:**

_Small suggestions that did not impact the score of the paper:_
- Figure legends would be improved if filenames were replaced by English phrases.
- L63-64: This sentence does not make sense to me. Perhaps you mean "EnvPool increases environmental throughput without affecting the learning algorithm"?
- L66: "in the previous literature" -> "in the literature"
- L115: Replacing "return" with "returned value" would make this sentence more clear, since "return" has a specific meaning in RL.
- Figure 2 could probably be placed side-by-side to reduce the space it takes up.
- L254: "we can search alternative..." -> "we can search an alternative..."
- L255: "PPO by default use..." -> "PPO by default uses..."
- L256: "re-use" -> "re-uses"


**Clarity:**

The paper is fairly well written and understandable, the only portion of the paper that was unclear was lines 205 - 214. What are these bullet points referencing? I have assumed that they refer to the choices made in experiment 1, but this is not clarified in the paper. Also, the paper refers to "workers" without defining what these are. I assume they are "CPUs".

**Correctness:**

The paper does not provide enough experimental details to fully understand the experiments that were run, nor to gauge the correctness of these experiments.  Furthermore, uncertainty of results is not discussed in detail.

For experiment 1 on pure environment simulation (section 4.1), the number of runs performed is not clear, and so I assume only a single run of the experiment was performed. Were multiple runs performed? If not, why was only one run used? Why not report mean performance improvement with some measure of uncertainty over many trials? Perhaps this would show the benefits of EnvPool are better or worse from what is shown here.

In experiment 1, the best performance of EnvPool (async) and Sample-Factory are reported, tuned over multiple `num_envs` given a constant `batch_size`.  Why was this choice made?
Why was a consistent `num_envs` not used for all execution engines in this experiment? Setting `num_env` for some execution engines while tuning it for others seems to bias the results in favour of those for which this hyperparameter was tuned.
Wouldn't using a constant `num_env` for all algorithms be a fairer comparison?
Perhaps I am not understanding the experimental details correctly. Can this experimental decision be qualified?

For experiment 2 on end-to-end agent training (section 4.2), multiple experimental details needed to gauge correctness are also lacking. How many runs were used to generate these results? What are shaded regions describing? How were PPO's hyperparameters tuned?  From the text, it sounds like hyperparameters were tuned to increase EnvPool's throughput, why was this choice made? Wouldn't this choice give the baseline execution engines a disadvantage?

The purpose of experiment 2 is to show that EnvPool can train algorithms faster by an increase in environment throughput. Unfortunately, this experiment may give EnvPool an advantage over the baselines execution engines. When running PPO on EnvPool, the paper clarifies that there will be a reduction of the usage of stale data (the definition of which is unclear).  This does not occur with the baseline engines. Perhaps the experiment would be better if this confounding factor was removed. Maybe the increase in performance (equivalently, the decrease in required training time) is not due to EnvPool but rather due to the data used by PPO. Furthermore, on line 256, what does it mean to reuse data with 10 epochs? Does this mean PPO is updated with the same data for 10 updates?

A number of experiments in the appendix use different underlying environments for each execution engine (MuJoCo v4 vs v2) and then compare the results. Why was this choice made? Wouldn't it be a more fair evaluation if both execution engines were run on the same environment?

**Documentation:**

EnvPool has extensive documentation.

**Ethics:**

No ethical concerns exist for this work.

**Relation To Prior Work:**

The paper does a satisfactory job in relating to prior work.


**Summary And Contributions:**

This work introduces EnvPool an open-source software for parallel environment execution. EnvPool aims to speed up the execution time of existing reinforcement learning (RL) benchmarks, which can be a bottleneck for performance.  EnvPool can be run in synchronous or asynchronous mode and run faster than similar software. EnvPool is implemented in C++ and custom environment can be implemented in C++.

---

> ### Author Response · Authors · 2022-08-09
> **EnvPool matches perfectly in the scope of the venue**
>
> From the [Call for paper website](https://neurips.cc/Conferences/2022/CallForDatasetsBenchmarks), the program chairs described the scope of the venue, which we quote as follows.
> > SCOPE. In addition to new datasets and benchmarks on new or existing datasets, we welcome submissions that detail advanced practices in data collection and curation that are of general interest even if the data itself cannot be shared. **Data generators**, **reinforcement learning environments**, or benchmarking tools are also in scope. Frameworks for responsible dataset development, audits of existing datasets, **identifying significant problems with existing datasets and their use**, or systematic analyses of existing systems on novel datasets that yield important new insight are also in scope.
>
> According to the description, our paper fits perfectly in the scope of the datasets and benchmarks track. EnvPool is a data generator that provides environment experiences for agents to interact and learn from. We have identified that the environment execution part is the common bottleneck of RL training systems and curated a highly parallel system to speed it up by multiple times.
>
> We could refer to publications from the [1st proceeding of this venue](https://nips.cc/Conferences/2021/DatasetsBenchmarks/AcceptedPapers/). Brax and IsaacGym were published in NeurIPS 2021 Datasets and Benchmarks. They share the same high-level motivation as our paper which is to speed up the simulation and leverage highly parallel systems.
>
> Finally, considering the paper is quite well received by all other reviewers, we do not think it’s inappropriate to have the paper in the datasets and benchmarks track. If you still have doubts about this, we should invite AC/Senior AC/PC to judge the criteria.
>
> Please evaluate our paper from a technical perspective instead of simply rejecting it due to a light doubt of the suitable venue. Hope the clarification helps!

---

> ### Author Response · Authors · 2022-08-10
> **Respose to Technical Questions**
>
> > Q: For experiment 2 on end-to-end agent training (section 4.2), multiple experimental details needed to gauge correctness are also lacking. How many runs were used to generate these results? What are shaded regions describing?
>
> We thank reviewers for this catch. Our results are reported over 5 random seeds. The shaded regions represent the standard error of the episodic returns over 5 random seeds.
>
> > Q: How were PPO's hyperparameters tuned?
>
> We are planning to rewrite the end-to-end experiment section to better identify the impact of environment execution in the training system and better explain why and how we are tuning the hyperparameters.
>
> Specifically, regarding the CleanRL + Atari experiments in Figure 4, we have used the same hyperparameters as in the original PPO paper (Table 5 in Schulman et al., (2017)). CleanRL’s hyperparameters in Figure 6 were tuned via [Weights and Biases’](https://wandb.ai/) automated hyperparameters search that optimizes average normalized scores in HalfCheetah-v4, Walker2d-v4, and Ant-v4 for 3 random seeds using Bayes optimization. CleanRL’s hyperparameters in Figure 7 were tuned via a similar procedure to optimize just for Pong-v5. The hyperparameters tunning process is tracked and the results are openly available. If you'd like to know the exact details we can provide additional links.
>
> All the hyperparameters used rl_games were tuned through trial and error, following the same practice in IsaacGym.
>
> Schulman, John, Filip Wolski, Prafulla Dhariwal, Alec Radford, and Oleg Klimov. "Proximal policy optimization algorithms." arXiv preprint arXiv:1707.06347 (2017).
>
> > Q: Furthermore, on line 256, what does it mean to reuse data with 10 epochs? Does this mean PPO is updated with the same data for 10 updates?
>
> Yes, it does. In MuJoCo, the openai/baselines’ PPO by default uses `N=1` and `num_steps=2048`, which collects 2,048 steps of data per rollout. It then updates the agent on these 2,048 steps of data for 10 epochs. In each epoch, it breaks these 2,048 steps of data into 32 mini-batches of size 64 and does a gradient pass for each mini-batch. So given the same 2,048 steps of data, openai/baselines’ PPO does 32 * 10 = 320 gradient updates. After PPO finishes the first gradient update, the agent’s policy is no longer the same as the behavior policy used for rollout, so PPO uses “stale data” for the remaining 319 gradient updates. Please refer to [ICLR blog post](https://iclr-blog-track.github.io/2022/03/25/ppo-implementation-details/) and https://arxiv.org/abs/2205.09123 for more details.
>
> > Q: From the text, it sounds like hyperparameters were tuned to increase EnvPool's throughput, why was this choice made?
>
> As mentioned above, openai/baselines’ PPO does 32*10 = 320 and only uses `N=1`, so envpool could not offer any benefit because there is nothing to parallelize when `N=1`. To better leverage EnvPool, we needed to search an alternative set of hyperparameters that uses a larger N like `N=64`.
> For example, the CleanRL’s tuned parameters use less stale data – we use `num_envs=64` and `num_steps=64`, which collects 4,096 steps of data per rollout, and use 2 epochs and 4 mini batches, which translates to 8 gradient updates in total compared to openai/baselines’ setting that uses 320 gradient updates.
>
> > Q: Maybe the increase in performance (equivalently, the decrease in required training time) is not due to EnvPool
> but rather due to the data used by PPO [...] Wouldn't this choice give the baseline execution engines a disadvantage?
>
> The increase in performance comes from tuned parameters. So if we run openai/baselines’ PPO with `N=64, num_steps=64`, 2 epochs and 4 mini batches, and the same other hyperparameters, we should expect to see an identical performance in 10M steps. [ICLR blog post](https://iclr-blog-track.github.io/2022/03/25/ppo-implementation-details/) has readily shown equivalence between openai/baselines’ PPO and CleanRL’s PPO.
>
> The issue is that if we run openai/baselines’ PPO (or CleanRL’s PPO) with `N=64` in MuJoCo using For-loop or Subprocess, the speed would be considerably slower — this is where EnvPool comes in and makes sure running with `N=64` is still fast.
>
> Sorry for the confusion, we plan to rewrite the high-throughput training section to better explain all these.

---

> > ### Comment · Reviewer_9AXU · 2022-08-10
> > **Response to Author Response**
> >
> > I thank the authors for the technical clarifications and additional in-text descriptions they have made, which have addressed my concerns. I have one remaining comment:
> >
> > In the newly added Figure 4, can clarification be made as to why EnvPool has only been used in a synchronous manner? I suspect EnvPool (async) would offer increased performance here. As it stands, although EnvPool (sync) outperforms `For-loop`, whether EnvPool (sync) outperforms Subprocess is unclear. I suspect EnvPool (async) would better display the benefits in this situation. A larger number of runs could also be conducted to increase the confidence in performance of execution engines in Figure 4 and confirm the benefits of EnvPool (sync) over Subprocess.

---

> > > ### Author Response · Authors · 2022-08-11
> > > **Response on the remaining comment:**
> > >
> > > We thank reviewer 9AXU for the prompt response, and we are glad that our response has addressed your concerns.
> > >
> > > > Q: In the newly added Figure 4, can clarification be made as to why EnvPool has only been used in a synchronous manner? I suspect EnvPool (async) would offer increased performance here.
> > >
> > > Yes absolutely. We used EnvPool (sync) because it provides an identical API as the Vectorized Envrionment API in gym (see [gym's docs](https://www.gymlibrary.ml/content/vector_api/)), so we can use EnvPool (sync) as a drop-in replacement. In general, EnvPool (sync) plays nicely with the learning algorithm (e.g., the standard PPO implementations).
> > >
> > > Working with EnvPool (async) is essentially working with an asynchronous PPO implementation, which is non-trivial. Specifically, EnvPool (async) introduces a series of challenges to the standard PPO implementations for at least two reasons.
> > > 1. Storage: batch of envs are returned in a first come first serve manner (see https://github.com/sail-sg/envpool#asynchronous-api) in no particular order, so PPO needs to store the `env_ids` of the batch of envs and retrieve the data properly during training.
> > > 2. GAE calculation: EnvPool (sync) makes sure the environment steps are synchronized, so if we have `num_steps=128` and N=4, then we know each environment executed 128 steps and we can write vectorized code to calculate Generalized Advantage Estimation (GAE) like `for t in reversed(range(num_steps)): delta = rewards[t]...`. With EnvPool (async), however, all bets are off — imagine env 1 steps really fast for some reason and we ended up with env1 stepping 400 steps, env2 stepping 100 steps, env3 stepping 6 steps, env4 stepping 6 steps. All of a sudden, the value function needs to fit a return calculated using up to 400 steps and 6 steps at the same time, which seems like a poor choice. We could wait for env2 to step an additional 300 steps in the next rollout phase, but then the first 100 steps is collected with a different policy for env2. The takeaway is utilizing EnvPool (async) completely changes the algorithm and it is a non-trivial effort to adopt it.
> > >
> > > We have some early prototypes to work with EnvPool (async) in a more restricted way that still ensures all envs step 128 steps (which I call “soft async”). Feel free to do a filediff between https://github.com/vwxyzjn/envpool-xla-cleanrl/blob/main/ppo_atari_envpool_soft_async_jax.py  and https://github.com/vwxyzjn/envpool-xla-cleanrl/blob/main/ppo_atari_envpool_jax.py. The amount of refactoring in the learning code is non-trivial.
> > >
> > >
> > >
> > > > Q: As it stands, although EnvPool (sync) outperforms For-loop, whether EnvPool (sync) outperforms Subprocess is unclear.  A larger number of runs could also be conducted to increase the confidence in the performance of execution engines in Figure 4 and confirm the benefits of EnvPool (sync) over Subprocess.
> > >
> > > Figure 4 shows the average time of 9765 iterations so it should offer concrete evidence. Unlike RL results, time results subject to significantly less stochasticity — see https://imgur.com/a/RYXBFE2 for how the average times change overtime. Note that EnvPool (sync) in Figure 4 is roughly 100% faster than Subprocess, which is consistent with our findings in Figure 3 (a). This further suggests we can rely on Figure 3 to develop expectations on the speed up in Environment Step Time.
> > >
> > > On a related note, we are planning to additionally do the ppo mujoco experiments that uses N=64 with EnvPool (sync) and Subprocess — this experiment should more clearly show EnvPool (sync)’s superiority (especially on a machine that only has 24 cores).

---

> > > > ### Comment · Reviewer_9AXU · 2022-08-17
> > > > **Response to the clarifications on the final comment.**
> > > >
> > > > I thank the authors for the clarification on the results of Figure 4. My concerns have been mostly addressed, and I will adjust my scoring accordingly.

---

> ### Author Response · Authors · 2022-08-10
> **Response to Technical Questions II**
>
> > Q: A number of experiments in the appendix use different underlying environments for each execution engine (MuJoCo v4 vs v2) and then compare the results. Why was this choice made? Wouldn't it be a more fair evaluation if both execution engines were run on the same environment?
>
> We did this mainly because v4 and v2 should offer comparable results based on months of benchmark and alignment effort (see this [Merge Request](https://github.com/openai/gym/pull/2762) on openai/gym). Another less exciting reason is related to engineering. The MuJoCo v4 only lives in `gym>=0.24.0`, which introduces **a series of breaking changes** that makes it difficult to run v4 experiments with CleanRL and openai/baselines. The recommendation from the gym maintainers is to migrate *until* they have finalized the API design and bug fixes in the upcoming months.

---

### Author Response · Authors · 2022-08-09
**Common questions shared by a few reviewers**

Here we put together some shared questions asked by reviewers.

> Q: User friendliness due to C++ codebase

We would like to clarify there are two groups of targeted users for EnvPool. One is RL researchers and practitioners who do not have to modify any parts of the RL environments. For example, researchers who would like to train an agent on Atari / Mujoco tasks. They can use EnvPool just as OpenAI Gym, but faster. EnvPool intends to cover as much as standard RL environments as possible in our GitHub repository. This group of users does not need to understand any internals of EnvPool, including any C++ code. They only work with the Python APIs.

The other group of “users” which we’d like to call developers, are familiar with RL environments implementation (in C++) and would like to integrate their loved RL environments into EnvPool to speed up the environment execution. For this developers group, we’ve provided extensive documentation on how to integrate a C++-based RL environment into EnvPool, including some simple examples. The process includes using Bazel build tools which might be unfamiliar to some developers but we’re always willing to help out. Python APIs will be finally exposed via pybind to researchers. Note that we do naturally have a higher engineering expectation on developers, they should be able to go through the integration process by reading the documentation and examples.

The concerns of using C++ hurting the adoption of EnvPool might come from misunderstanding and confusion between the two targeted groups. Hope that the clarification above helps!

> Q: Comparisons with GPU-accelerated environments like Brax and IsaacGym

**Targeted environments are different.** As we mentioned in the paper, only a small number of RL environments (where matrix operations are the key of the environments) can be reimplemented into GPU accelerated environments like Brax and IsaacGym. Modern video games, real-world RL simulators, etc can only be run on CPUs for simulation. Users should just choose whatever execution engine fits their research and application requirements.

**Fair comparisons are not easy.** We should also note that fair comparisons are not easy to make here. For example, to compare with IsaacGym on an NVIDIA A100 GPU, how many CPUs should we utilize for EnvPool? Another aspect is that, though Brax, IsaacGym, and Mujoco (with EnvPool) are all Mujoco-like simulators, they use different internals (e.g. collision system) for their simulations.

> Q: Analysis of impact of environment execution to the whole training system

This is a great question! Thank you. To address this question, we added an analysis section in the end-to-end training titled “​​How much time does parallel environment execution take?”. Apologies for the inconsistency with image styles — we will fix them later.

---

### Meta-Review · Area_Chair_M9Ub · 2022-09-08

**Recommendation:** Accept
**Confidence:** 4

**Metareview:**

I would like to thank the authors for their submission and for addressing the reviewers' comments. I also want to thank the reviewers for their helpful feedback.

The paper introduces EnvPool, an open-source system that parallelizes the execution of reinforcement learning workloads on a single node. The code is well documented and is supported on GitHub. Some reviewers were able to replicate the results of the pure environment benchmarking using documented code. This work is very important, especially due to improving the rollout efficiency, which is a pain point for RL researchers that often spend days waiting for their RL training to finish. The authors do a good job of addressing the majority of the reviewer's concerns and incorporating these changes in the final draft.

Main Pros of EnvPool:
* Has support for some of the widely used simulation systems like Atari/MuJoCo and can be used off the shelf.
* Speeds up the execution of RL benchmarks, which speeds up the research cycle.
* Has a Python API, which is simple to use and can be integrated easily into existing RL training setups.

Main Cons of EnvPool:
* It only supports Single node execution. With bigger and more complex models, a large portion of RL workloads is running on multiple machines.
* Supporting a custom RL environment requires writing C++ code which is not user-friendly.
* In general, this work seems to be targeted at the case where the environment is extremely cheap to simulate. Such simulators are not very common outside of research. It would be great if the authors explain how (or if) this work can be applied to more complex environments that cannot be simulated.

Considering the pros and cons above and the reviewers' insightful feedback, I recommend accepting this paper and presenting it as a poster in the program. Congratulations!

---

### Decision · Program_Chairs · 2022-09-16

Accept